# Assessment of the Effects of Commercial or Locally Engineered Biochars Produced from Different Biomass Sources and Differing in Their Physical and Chemical Properties on Rumen Fermentation and Methane Production In Vitro

**DOI:** 10.3390/ani13203280

**Published:** 2023-10-20

**Authors:** Chaouki Benchaar, Fadi Hassanat, Cristiano Côrtes

**Affiliations:** 1Agriculture and Agri-Food Canada, Sherbrooke Research and Development Centre, Sherbrooke, QC J1M 0C8, Canada; 2Agriculture and Agri-Food Canada, Quebec Research and Development Centre, Quebec, QC G1V 2J3, Canada; fadi.hassanat@agr.gc.ca; 3Agrinova, Alma, QC G8B 7S8, Canada; cristiano.cortes@agrinova.qc.ca

**Keywords:** biochar, methane production, in vitro

## Abstract

**Simple Summary:**

This in vitro study was undertaken to assess the effects of seven biochars (four commercial and three locally engineered) on rumen microbial fermentation and methane production. The biochars were produced via the pyrolysis of different biomass sources and differed in their chemical and physical properties. They were evaluated at 1%, 2%, or 5% of the substrate’s dry matter using batch cultures of ruminal fluid (with 24 h incubation). Despite the contrasting physical and chemical characteristics of the biochars evaluated, neither rumen fermentation (pH, volatile fatty acids, and ammonia-nitrogen) nor methane production were affected. At the examined doses and under the experimental conditions, biochar was ineffective, and it is not a viable option for mitigating enteric methane production.

**Abstract:**

In recent years, interest in using biochar as feed additives to mitigate enteric methane (CH_4_) emissions from ruminants has increased. It has been suggested that the mitigating potential of biochar is influenced by its physical (e.g., porosity-related) and chemical (e.g., redox-potential-related) properties. Thus, the aim of this in vitro study was to evaluate the effects of commercial or locally engineered biochars, produced from different biomass sources and differing in their physical and chemical characteristics, on rumen fermentation and CH_4_ production. For this purpose, a 24 h batch culture of ruminal fluid incubations was conducted in a complete randomized block design (repeated three times) that included a negative control (no additive), a positive control (monensin, 10 mg/mL), and four commercial and three locally engineered biochars, each evaluated at 1%, 2%, or 5% of the substrate’s (i.e., the total mixed ration) dry matter. The evaluated biochars greatly differ in their chemical (i.e., moisture, ash, pH, redox potential, volatiles, carbon, fixed carbon, hydrogen, and sulfur) and physical (i.e., fine particles < 250 µm, bulk density, true density, porosity, electrical conductivity, specific surface area, and absorbed CO_2_) properties. Despite these differences and compared with the negative control, none of the biochars evaluated (regardless of the inclusion rate) influenced gas and CH_4_ production, volatile fatty acid characteristics (total concentration and profile), or ammonia-nitrogen (NH_3_-N) concentrations. As expected, monensin (i.e., the positive control) decreased (*p* < 0.05) CH_4_ production mainly because of a decreased (*p* < 0.05) acetate-to-propionate ratio. The results of this study reveal that despite the large differences in the physical and chemical properties of the biochars evaluated, their inclusion at different rates in vitro failed to modify rumen fermentation and decrease CH_4_ production. Based on these in vitro findings, it was concluded that biochar does not represent a viable strategy for mitigating enteric CH_4_ emissions.

## 1. Introduction

Several feed additives have been assessed for their effects on enteric methane (CH_4_) production in dairy cows [1,2]. For instance, monensin, an ionophore antibiotic, has been shown to decrease enteric CH_4_ production, although its inhibitory effect does not persist over time [3], most likely because of the capacity of rumen microbes (i.e., protozoa) to adapt to monensin exposure in the rumen [4]. On the other hand, in recent years, public concern over the routine use of feed antibiotics and chemical additives in livestock nutrition has increased due to the residues potentially transferable to animal products [5,6]. Therefore, there is a great deal of interest in developing alternatives to this type of feed additive to promote efficient feed utilization while reducing the impact of livestock production on the environment.

More recently, interest in using biochar in ruminant nutrition has increased because it has been suggested to be a potential means of reducing enteric CH_4_ production [1,2,7]. Biochar has shown effectiveness in reducing CH_4_ emissions in soils [8,9,10] and compost [11,12]. These findings have encouraged a plethora of animal scientists to investigate whether the anti-methanogenic properties of biochar can be exploited to inhibit ruminal methanogenesis. By definition, biochar is a carbon-rich product obtained through the thermal decomposition of various sources of biomass (e.g., animal wastes, plant residues, and lignocellulosic plant materials) under a limited supply of oxygen at temperatures ranging from 350 to 1000 °C [13,14]. This production process, known as pyrolysis, generates an extremely porous, high-surface-area material that is bioactive and binds organic compounds. The properties of biochar vary greatly depending on the nature of the organic material and the conditions of partial pyrolysis [15].

For many centuries, biochar has been used to treat digestive disorders in both in humans and livestock [16]. Biochar has also been successfully used to amend soils and increase their nutrient availability beyond a fertilizer effect [17]. It has been proposed that the application of biochar to soil reduces CH_4_ emissions by increasing soil aeration through promoting the activities of the methanotrophic population and increasing the population ratio of methanotrophs to methanogens [18]. It is only since 2010 that biochar has been increasingly used as a feed additive in livestock production [19], in particular to mitigate enteric CH_4_ emissions from ruminants. Different hypotheses have been suggested to explain the mechanisms involved in the reduction in gas and CH_4_ production through biochar supplementation. Nevertheless, it appears that the main reason is related to the ability of biochar to absorb and adsorb gases [17,20].

A number of studies, most of them in vitro, have assessed the effects of biochar on enteric CH_4_ production. The in vitro CH_4_ responses to biochar supply varied from no effect [21,22] to a decrease [23,24,25,26]. An important part of this variation is mainly related to the source of biomass used to produce the biochar and the physicochemical characteristics of the biochar, as influenced by the pyrolysis process [15,27]. Despite the number of studies published to date, the range of biomasses that can be used to produce biochar is extensive, and different biomass sources may affect the quality and the composition of a biochar, thus affecting the anti-methanogenic capacity of the biochar. In addition, there still is no clear indication of the specific physicochemical characteristics of biochars that make them effective in inhibiting ruminal methanogenesis. Accordingly, the objective of this study was to assess the effects of different sources of commercially available or locally engineered biochar on rumen microbial fermentation and CH_4_ production in vitro.

## 2. Materials and Methods

### 2.1. Rumen Inoculum

Ruminal contents were collected from two ruminally cannulated lactating Holstein cows fed a total mixed ration (TMR) consisting of 50% forage and 50% concentrate (Table 1). The TMR was formulated to meet or slightly exceed the nutrient requirements of the cows [28] and was offered twice daily (0900 and 1600) for ad libitum intake. Cows were cared for in accordance with the guidelines of the Canadian Council on Animal Care [29].

Rumen contents were collected before the morning feeding from the anterior dorsal, anterior ventral, medium ventral, posterior dorsal, and posterior ventral locations within the rumen. The rumen contents were placed in an insulated thermos and transported immediately to the laboratory, where they were homogenized using a mixer and strained through two layers of cheesecloth into a pre-warmed (39 °C) bottle. The strained rumen fluid was combined with medium of Menke et al. [30] in a ratio of 1:5. The buffered rumen fluid was purged continuously under free-oxygen CO_2_ and kept at 39 °C in a water bath prior to use for in vitro incubations.

### 2.2. Substrate

A representative sample of the TMR fed to the donor cows was freeze-dried and ground to be able to pass through a 1 mm screen using a Wiley mill (Standard Model 4, Arthur M Thomas, Philadelphia, PA, USA) for later use as in vitro fermentation substrate.

### 2.3. Experimental Treatments

The experiment was conducted as a complete randomized block design and was repeated three times on three separate days (1 week apart). Evaluated biochars were (1) Happy Tummy (Fine Fettle Products; Narberth, UK), produced from rice husks; (2) Fine Premium (High Plains Biochar, LLC, Laramie, WY, USA), produced from whole pine trees, including limbs and needles; (3) Carbon 2M (Titan, Craik, SK, Canada), produced from shredded and ground forestry wood; (4) Airex (Airex, Bécancour, QC, Canada), produced from a mixture of forest biomass; (5) Roasted pellets (Biochar Boréalis-Agrinova, Mashteuiatsh, QC, Canada), produced from wood pellets made from spruce shavings; (6) Local-R (Biochar Boréalis-Agrinova, Mashteuiatsh, QC, Canada), produced from sieved black shavings; and (7) Local-L (Biochar Boréalis-Agrinova, Mashteuiatsh, QC, Canada), produced from black spruce shavings.

The engineered (i.e., locally produced) biochars (i.e., roasted pellets, Local-R, and Local-L) were produced by pyrolyzing the biomass sources using commercial biomass carbonization equipment (Biogreen technology; ETIA S.A.S, Compiègne, France). The temperatures and the durations of pyrolysis were 450 °C, 500 °C, and 450 °C and 30 min, 15 min, and 30 min for roasted pellets, Local-R, and Local-L, respectively. 

Each biochar was added at concentrations of 1%, 2%, and 5% of substrate dry matter (DM). When expressed as milligrams per liter of ruminal fluid culture, the concentrations of 1%, 2% and 5% of substrate DM correspond to 100 mg/L, 200 mg/L, and 500 mg/L, respectively. These dose rates are within the range (i.e., 1% to 5% of substrate DM) of those reported in other previous in vitro studies [21,22,26,31]. In addition, negative (CTL, no additive) and positive (Monensin, MON, 10 mg/L; Sigma Aldrich, St. Louis, MO, USA) controls were also included in the incubations. All treatments (including blank and control) were evaluated in triplicate (i.e., *n* = 3 bottles) in the same run (i.e., the same day). The experimental unit was the average of bottles (*n* = 3) within the run (i.e., day), which provided three experimental units for each experimental treatment.

### 2.4. In Vitro Incubation

In vitro incubations were performed in 100 mL pre-warmed (39 °C) serum bottles containing 200 mg of the substrate (DM basis) and the experimental treatments. To begin the incubations, 20 mL of buffered rumen fluid was dispensed into each bottle and purged continuously with O_2_-free CO_2_. Blank bottles (*n* = 3) containing buffered ruminal fluid only were also included in each run. The bottles were sealed with butyl rubber stoppers and aluminum PTFE Teflon seals and placed in a water bath set to shake at 50 rpm at 39 °C for 24 h.

At the end of incubation, the bottles were taken out of the water bath. The pressure of gas produced was measured using a pressure transducer digital pressure gauge (Ashcroft 2089, Ashcroft Inc., Stratford, CT, USA) and used to calculate the volume of gas produced according to the following equation:P_1_ × V_1_ = P_2_ × V_2_

Here, P_1_ = pressure of the incubation chamber, V_1_ = volume of gas produced at atmospheric pressure, P_2_ = pressure of gas measured at specific incubation time, and V_2_ = volume of headspace of the bottle. The gas produced was sampled for CH4 measurement. After gas production measurement and sampling, the pH of the buffered rumen fluid was measured (Orion Star A211 pH meter, Thermo Scientific, Beverly, MA, USA), and the bottles were immediately immersed in an ice bath to impede microbial activity. The buffered rumen fluid was then centrifuged at 22,000× *g* for 20 min to separate solids from the liquid phase. The liquid portion was sampled for later analyses of ammonia-nitrogen (NH_3_-N) and volatile fatty acid (VFA) concentrations.

### 2.5. Chemical Analyses

The dry matter, organic matter, crude protein, and starch content of the TMR was analyzed according to the AOAC [32]. Methane was analyzed using a 490 Micro GC Biogas analyzer (Agilent technologies, Amstelveen, North Holland, The Netherlands) equipped with a 10 m PPQ column (Agilent technologies, Amstelveen, North Holland, The Netherlands) and a thermal conductivity detector. The column was operated at a carrier gas (He) pressure of 20 PSI, and temperature was fixed at 70 °C, while injector and detector temperatures were fixed at 110 °C and 70 °C, respectively. About 7 mL of the gas sample was injected manually in the inlet, and total run time was 1 min. Calibrations were performed daily using standard gas mixtures containing different proportions of CH_4_. Analysis of VFA was performed using a gas chromatograph equipped with a flame ionization detector and auto-injector (6850 network GC system, Agilent technologies, Mississauga, ON, Canada) fitted with a DB-FFAP column (30 m × 0.250 mm × 0.25 µm; Agilent technologies, Mississauga, ON, Canada). Ammonia-N concentration was determined as reported in the study conducted by Weatherburn [33]. The chemical (i.e., moisture, ash, pH, redox potential, volatiles, carbon, fixed carbon, hydrogen, and sulfur) and physical (i.e., fine particles < 250 µm, bulk density, true density, porosity, electrical conductivity, specific surface area, and absorbed CO_2_) characterizations of the biochars were carried out according to the methods described by Singh et al. [34].

### 2.6. Statistical Analyses

Data were analyzed as a randomized complete block design using the mixed procedure of SAS (SAS Institute Inc., Cary, NC, USA) according to the following model:*Yij* = *µ* + *Ti* + *Dj* + *eij*

Here, *Yij* is the observation in treatment *i* on day *j*, *µ* is the overall mean, *Ti* is the fixed effect of treatment, *Dj* is the random effect of day, and *eij* is the error term. Differences between least square means of the CTLs (i.e., negative) and treatments (i.e., MON and biochars) were declared significant (*p* ≤ 0.05) using Dunnett’s test.

## 3. Results

### 3.1. Physical and Chemical Characteristics of the Biochars

Data on the physical and chemical characteristics of the evaluated biochars are presented in Table 2. These properties varied widely among the different sources of biochar. The moisture content ranged from 0.89% for Fine Premium to 9.13% for Airex, whereas the ash concentration varied from 1.10% for roasted pellets to 34.6% for Happy Tummy. The pH variation ranged from 6.20 (Airex) to 8.40 (Fine Premium). Extreme variation in the redox potential was observed, as the values ranged from −83.1 mV to 41.4 mV for Local-R and Airex, respectively. A wide magnitude of variation was also noted for volatiles, the values for which varied from 14.4% of DM for Local-R to 39.2% of DM for Fine Premium. Likewise, the concentrations of carbon and fixed carbon also varied with the source of biochar, with maximums of 88.5% of DM (Local-R) and 83.8% of DM (Carbon 2M) and minimums of 70.4% of DM (Happy Tummy) and 34.5% of DM (Happy Tummy) for carbon and carbon-fixed, respectively. Large variations in hydrogen, nitrogen, and sulfur concentrations were also observed. The H_2_ concentration ranged from 1.50% of DM (Fine Premium) to 3.54% of DM (Airex). The maximum and minimum values of N concentration were observed in Happy Tummy (1.27% of DM) and roasted pellets (0.27% of DM). The concentration of sulfur varied from 0.033% of DM in Carbon 2M to 0.586% of DM in Happy Tummy.

A large variation was also observed in the physical characteristics of the assessed biochars. The proportion of fine particles (i.e., <250 µm) varied extremely, ranging from a maximum of 56.8% of DM in Airex to 2.35% of DM in Happy Tummy. Similarly, the bulk and true densities varied widely with the source of the biochar. The minimum and maximum values for bulk density were 73 kg/m^3^ in Fine Premium and 434 kg/m^3^ in Happy Tummy. True density ranged from 1440 (Airex) to 1942 kg/cm^3^ (Fine Premium). The porosity values varied from 0.72 cm^3^/g in the roasted pellets to 0.96 cm^3^/g in Fine Premium, whereas the electrical conductivity varied from 179 µS/cm in Local-R to 406 µS/cm in Fine Premium. For specific surface area, the minimum and maximum values were 9.57 m^2^/g in Carbon 2M and 505 m^2^/g in Fine Premium. The concentration of absorbed CO_2_ varied from 0.65 mmol/g in Happy Tummy to 2.72 mmol/g in Fine Premium.

### 3.2. Effects of the Experimental Treatments

Data on the effects of the experimental treatments on gas production, rumen microbial fermentation characteristics, and CH_4_ production are presented in Table 3, Table 4, Table 5, Table 6, Table 7, Table 8 and Table 9. Compared with the CTL, the addition of MON (i.e., positive control) at 10 mg/L decreased (*p* < 0.05) the production of gas (35.9 vs. 46.7 m*M*) and CH_4_ (6.51 vs. 8.80 m*M*). In contrast, there were no changes (*p* > 0.05) in media pH and total VFA concentration. Supplying MON decreased (*p* < 0005) acetate and butyrate molar proportions but increased (*p* < 0.05) propionate molar proportions. As a consequence, the acetate/propionate ratio was lower (*p* < 0.05) for MON than for the CTL. Likewise, the branched-chain volatile fatty acid (BCVFA) molar proportion decreased (*p* < 0.05) with the addition of MON compared to the CTL, but no change in NH_3_-N concentration was observed, although the concentration was numerically lower for MON than for the CTL (14.2 vs. 16.9 md/dL). Contrary to MON, the addition of different sources of biochar (i.e., Happy Tummy, Fine Premium, Carbon 2M, Airex, roasted pellets, Local-R, and Local-L) at doses of 1%, 2%, or 5% of substrate DM had no effect (*p* > 0.05) on the production of gas and CH_4_, media pH, VFA (total and molar proportions), and NH_3_-N concentration.

## 4. Discussion

In the current study, MON was used as a positive control. The monensin ionophore has been reported to decrease CH_4_ production both in vitro [35,36] and in vivo [37,38]. Monensin inhibits ruminal methanogenesis mainly by directing ruminal VFA patterns toward increasing propionate production at the expense of acetate production [2,39]. This kind of shift drives H_2_ utilization away from methanogenesis to propionogenesis given the inverse relationship between the two rumen fermentation processes. Such changes in rumen VFA occurred in the current study, in which the molar proportion of acetate decreased, whereas that of propionate increased with the addition of MON compared with CTL. As a consequence, CH_4_ production decreased by 26% compared with the CTL. A similar shift in VFA pattern (i.e., a lower acetate/propionate ratio) with the concurrent inhibition of CH_4_ production was also observed in a previous study conducted in our laboratory [35] when MON (10 mg/L) was compared with a CTL in 24 h batch-culture incubations.

This in vitro experiment examined biochars made from different biomass materials and differing in their chemical and physical properties. This target was achieved, as shown by the wide variation in both the physical and chemical properties. The physical characteristics that seem to play an important role in reducing CH_4_ production in the rumen are the porosity and specific surface area of the biochar. In this regard, it has been speculated that high specific surface area and porosity could favor the growth of methanotrophic bacteria in the rumen and facilitate anaerobic methane oxidation [7,40]. Accordingly, several industrial processes have been developed to increase the porosity and specific surface area of biochar [17].

In the present study, the different sources of biochar evaluated at incremental doses (i.e., 1%, 2%, and 5% of substrate DM) failed to alter rumen fermentation VFA (i.e., decreased acetate and increased propionate molar proportions) and reduce CH_4_ production. In vitro responses of CH_4_ production to biochar supplementation have been inconsistent in in vitro studies, with responses ranging from no effects [41,42,43] to decreases [23,24,25,26,31]. Such discrepancies between studies may be related to different factors, but it was suggested that the difference in the plants and/or biomass used and the heating temperature applied during the pyrolysis process were the main reasons. It has been suggested that these factors affect the adsorptive capacity of biochar and hence its potentially gaseous emissions [17,27,44]. Cabeza et al. [44] assessed the effect of biochar (10 or 100 g biochar/kg substrate fresh weight), produced from different biomass sources (Miscanthus straw, oil seed rape straw, rice husk, soft wood pellets, or wheat straw) and at two process temperatures (550 °C or 700 °C), on CH_4_ production in vitro using a total mixed ration (50% hay, 40% barley, and 10% rapeseed meal). The authors observed no change in CH_4_ production regardless of the inclusion rate, the biomass source, or the process temperature. In a more recent study, Tamayao et al. [22] determined the effect of three different pine-based biochars (included at the rate of 20 g/kg diet DM) with differing physicochemical properties on CH_4_ production in an artificial rumen (RUSITEC) fed barley silage.

Despite the differences in the physical (i.e., bulk density, specific surface area, and porosity) and chemical (i.e., carbon, fixed carbon, ash, and pH) characteristics of the biochars, no effects were observed on CH_4_ production. Contrasting results from the same research group were reported in a previous study by Saleem et al. [26], who assessed the effect of pine-based biochar in a RUSITEC system fed a high-forage (barley silage) diet. In that study, CH_4_ production (mg/d and g/g of DM incubated; g/g of DM digested) decreased quadratically when pine-based biochar was supplied at rates of 0.5%, 1%, and 2% of substrate DM. Compared with the control, the addition of biochar in the RUSITEC decreased CH_4_ production by 25%, depending on the unit of CH_4_ production expression. The difference in CH_4_ production responses to pine-based biochars between the study by Saleem et al. [26] versus that by Tamayao et al. [22] was likely due to the observed increase in DM disappearance in the study by Saleem et al. [26], which did not occur in the study by Tamayao et al. [22].

In the current study, despite large differences in the chemical and physical characteristics of the biochar and the plant/biomass used for its production, CH_4_ production was not changed regardless of the inclusion rate of the biochar. Such a lack of effects is therefore in line with the findings of Tamayao et al. [21,22] but in contrast with the results of Saleem et al. [26]. Therefore, variations in the effect of biochar on CH_4_ production among in vitro studies cannot be imputed to differences in the physicochemical characteristics of the biochar, and the reasons for such discrepancies remain unclear.

All the biochars evaluated at increasing doses did not affect total VFA concentrations or the amounts of gas produced, suggesting that the extent of substrate fermentation was unchanged. A lack of effect of biochar on nutrient disappearance in vitro was also observed by others [21,22,43]. In contrast, other in vitro studies reported increased total VFA production, which was consistent with the observed increase in DM degradability [26,42]. In our study, the different sources of biochars tested induced neither a reduction in acetate molar proportion nor an increase in that of propionate. This lack of change was reflected by the unchanged CH_4_ production between the control and the different sources of biochars supplied at increasing doses. Our VFA profile findings are in agreement with those of Mirheidari et al. [31] and Tamayao et al. [21,22] but contradict the results of Saleem et al. [26], who observed linear increases in daily production of acetate, propionate and BCVFA with increasing inclusion rates (5, 10, or 20 g of biochar/kg of substrate DM) of jack-pine-based biochar. In the study by Saleem et al. [26], the increase in acetate production was consistent with the observed enhancement in fiber digestion. However, propionate production was also increased in the study by Saleem et al. [26], resulting in stagnation in the acetate/propionate ratio, which is not consistent with the decrease in CH_4_ production observed in their study. Saleem et al. [26] speculated that an increase in methanotrophs might have caused the observed decline in CH_4_ production in vitro.

In the present study, NH_3_-N concentrations were not affected by the addition of biochars regardless of the level of inclusion of the biochar, which was consistent with the complete lack of an effect of biochar on BCVFA. Branched-chain VFA are the end-products of amino acid (AA) deamination in the rumen. Data in the literature reveal that the effect of biochar on ammonia production/concentration is very variable, as studies have reported no effect [21,22,42], a decrease [44], or an increase [26]. In the study by Cabeza et al. [44], the observed reduction in ammonia was explained by the possible adsorption of NH_3_-N by the biochar, a phenomenon that occurs in soils as biochars are used to prevent NH_3_-N leaching [45]. The prevention of such leaching may attenuate negative effects on the environment via the mitigation of greenhouse gas emissions, such as emissions of nitrous oxide [16]. Our results contrast with those of Ding et al. [45], who observed an increase in NH_3_-N concentration and BCVFA production with an increasing inclusion of biochar (0.5% to 2.0% of diet DM). In the study by Saleem et al. [26], concentrations of peptides (large and small) and AA were not changed by biochar addition, which may be an indication that biochar did not affect any step of the proteolysis process in the rumen. Thus, in general, it seems that the adsorptive property of biochar described in soils does not occur in the rumen, most likely because of the differences between the two ecosystems (e.g., greater concentrations and a shorter residence time in the rumen vs. in soils).

## 5. Conclusions

The results of this study show that despite large differences in the chemical and physical properties of various sources of biochars (commercially available or locally manufactured) produced from different biomass sources and under pyrolysis conditions, no changes occurred in rumen fermentation characteristics (pH, VFA, NH_3_-N) and CH_4_ production. Based on the experimental conditions of this study (24 h batch cultures; inclusion rates of 1%, 2%, and 5%), the biomass source used to produce the biochar, the pyrolysis conditions, and hence the physicochemical properties of the material produced, biochar was found to be ineffective in reducing CH_4_ production and therefore does not represent a viable option for dietary CH_4_ mitigation.

## Figures and Tables

**Table 1 animals-13-03280-t001:** Ingredients and chemical compositions of the total mixed ration fed to donor cows.

Item	% of Dry Matter
**Ingredient**	
Corn silage	40.0
Alfalfa silage	38.8
Soybean meal, 48% solvent extracted	7.74
Protein supplement ^a^	5.81
Corn grain, ground	3.76
Timothy hay, chopped	1.91
Minerals and vitamins supplement ^b^	1.49
Calcium carbonate	0.86
**Chemical composition**	
Dry matter	45.4
Organic matter	92.1
Crude protein	16.9
Neutral detergent fibre	32.3
Starch	15.5

^a^ Top supplement^®^, composed of 30% corn gluten meal, 20% heat-treated soybean, 20% canola meal, and 30% dried corn distillers grains (Bélisle Solution Nutrition Inc., St-Mathias, QC, Canada). ^b^ Contained 12.48% Ca, 6.80% P, 6.81% S, 7.72% Na, 1.97% K, 96 mg/kg I, 2877 mg/kg Fe, 620 mg/kg Cu, 2520 mg/kg Mn, 3777 mg/kg Zn, 83 mg/kg Co, 628,000 IU/kg vitamin A, 81,000 IU/kg vitamin D, 3739 IU/kg vitamin E, and 27.8 mg/kg Se.

**Table 2 animals-13-03280-t002:** Physical and chemical characteristics of the evaluated biochars.

	Biochar						
Item	Happy Tummy	Fine Premium	Carbon 2M	Airex	Roasted Pellets	Local-R	Local-L
**Chemical properties**							
Moisture (%)	5.94	0.89	1.76	9.13	3.13	1.59	4.35
Ash (% DM)	34.6	4.60	1.48	2.15	1.10	2.41	2.17
pH	6.25	8.40	7.30	6.20	7.85	7.30	7.70
Redox potential (mV)	33.8	0.00	−21.4	41.4	0.00	−83.1	0.00
Volatiles (% DM)	31.0	39.2	14.8	30.2	17.7	14.4	18.4
Carbon (% DM)	70.4	79.6	86.2	75.0	86.5	88.5	86.5
Fixed Carbon (% DM)	34.5	56.2	83.8	67.7	81.2	83.2	79.4
Hydrogen (% DM)	2.50	1.50	2.75	3.54	2.95	2.66	2.98
Nitrogen (% DM)	1.27	0.68	0.42	0.46	0.27	0.40	0.38
Sulfur (% DM)	0.586	0.317	0.033	0.042	0.055	0.034	0.035
**Physical properties**							
Fine particles < 250 µm (% DM)	2.35	15.3	2.41	56.8	5.01	14.8	24.4
Bulk density (kg/m^3^)	434	73	252	243	419	140	163
True density (kg/m^3^)	1623	1942	1485	1440	1474	1474	1498
Porosity (cm^3^/g)	0.73	0.96	0.83	0.83	0.72	0.91	0.89
Electrical conductivity (µS/cm)	357	406	226	223	229	179	233
Specific surface area (m^2^/g)	44.3	505	9.57	44.2	ND *	226	240
Absorbed CO_2_ (mmol/g)	0.65	2.72	1.98	1.50	2.26	2.25	2.30

* ND = not determined.

**Table 3 animals-13-03280-t003:** Effect of Happy Tummy biochar on rumen microbial fermentation and CH_4_ production in vitro.

			Biochar (% of DM)			
Item	CTL ^a^	MON ^b^	1%	2%	5%	SEM
Gas production (mL)	46.7	35.9 *	48.5	48.0	47.3	1.14
CH_4_ (mL)	8.80	6.51 *	8.70	8.62	8.75	0.123
pH	6.61	6.64	6.58	6.57	6.59	0.032
Total VFA (mM)	87.3	78.5	84.6	74.4	84.1	8.31
VFA (mol/100 mol)						
Acetate (A)	62.7	60.6 *	62.7	62.6	62.6	0.46
Propionate (P)	20.0	23.8 *	20.1	20.1	20.1	0.32
Butyrate	11.9	10.4 *	11.9	12.0	11.9	0.21
Valerate	1.74	2.03 *	1.74	1.74	1.74	0.059
BCVFA ^c^	3.64	3.20 *	3.62	3.65	3.63	0.086
A:P	3.13	2.56 *	3.12	3.12	3.11	0.057
NH_3_-N (mg/100 mL)	16.9	14.2	16.1	18.4	20.1	1.88

* Different from the control (CTL, 0 mg/L); *p* ≤ 0.05. ^a^ Control (0 mg/L). ^b^ Monensin (10 mg/L). ^c^ BCVFA: Branched-chain VFA = iso-butyrate + iso-valerate.

**Table 4 animals-13-03280-t004:** Effect of Fine Premium biochar on rumen microbial fermentation and CH_4_ production in vitro.

			Biochar (% of DM)			
Item	CTL ^a^	MON ^b^	1%	2%	5%	SEM
Gas production (mL)	46.7	35.9 *	48.0	47.2	49.1	1.00
CH_4_ (mL)	8.80	6.51 *	8.75	8.30 *	8.86	0.101
pH	6.61	6.64	6.60	6.59	6.60	0.034
Total VFA (m*M*)	87.3	78.5	71.8	83.5	80.7	6.25
VFA (mol/100 mol)						
Acetate (A)	62.7	60.6 *	62.7	62.9	62.7	0.47
Propionate (P)	20.0	23.8 *	20.0	20.0	20.2	0.30
Butyrate	11.9	10.4 *	11.9	11.8	11.9	0.25
Valerate	1.74	2.03 *	1.73	1.70	1.71	0.055
BCVFA ^c^	3.64	3.20 *	3.68	3.61	3.65	0.091
A:P	3.13	2.56	3.13	3.14	3.13	0.053
NH_3_-N (mg/100 mL)	16.9	14.2	17.4	16.9	15.3	1.66

* Different from the control (CTL, 0 mg/L); *p* ≤ 0.05. ^a^ Control (0 mg/L). ^b^ Monensin (10 mg/L). ^c^ BCVFA: Branched-chain VFA = iso-butyrate + iso-valerate.

**Table 5 animals-13-03280-t005:** Effect of Carbon 2M biochar on rumen microbial fermentation and CH_4_ production in vitro.

			Biochar (% of DM)			
Item	CTL ^a^	MON ^b^	1%	2%	5%	SEM
Gas production (mL)	46.7	35.9 *	47.6	48.0	47.8	0.968
CH_4_ (mL)	8.80	6.51 *	8.46	8.63	8.62	0.125
pH	6.61	6.64	6.59	6.59	6.58	0.031
Total VFA (m*M*)	87.3	78.5	81.0	75.9	78.2	11.31
VFA (mol/100 mol)						
Acetate (A)	62.7	60.6 *	62.7	62.5	62.7	0.52
Propionate	20.0	23.8 *	20.1	20.2	20.1	0.31
Butyrate	11.9	10.4 *	11.8	11.9	11.8	0.21
Valerate	1.74	2.03 *	1.72	1.73	1.72	0.065
BCVFA ^c^	3.64	3.20 *	3.58	3.64	3.62	0.105
A:P	3.13	2.56 *	3.12	3.09	3.11	0.059
NH_3_-N (mg/100 mL)	16.9	14.2	12.9	15.7	16.1	2.38

* Different from the control (CTL, 0 mg/L); *p* ≤ 0.05. ^a^ Control (0 mg/L). ^b^ Monensin (10 mg/L). ^c^ BCVFA: Branched-chain VFA = iso-butyrate + iso-valerate.

**Table 6 animals-13-03280-t006:** Effect of Airex biochar on rumen microbial fermentation and CH_4_ production in vitro.

			Biochar (% of DM)			
	CTL ^a^	MON ^b^	1%	2%	5%	SEM
Gas production (mL)	46.7	35.9 *	48.6	48.3	49.1	1.06
CH_4_ (mL)	8.80	6.51 *	8.72	8.80	8.41	0.100
pH	6.61	6.64	6.60	6.60	6.58	0.038
Total VFA (m*M*)	87.3	78.5	79.0	80.4	78.2	5.78
VFA (mol/100 mol)						
Acetate (A)	62.7	60.6 *	62.2	62.7	62.9	0.43
Propionate (P)	20.0	23.8 *	20.1	20.0	19.9	0.30
Butyrate	11.9	10.4 *	12.1	11.9	11.7	0.20
Valerate	1.74	2.03 *	1.79	1.73	1.87	0.097
BCVFA ^c^	3.64	3.20 *	3.74	3.65	3.59	0.086
A:P	3.13	2.56 *	3.09	3.13	3.16	0.052
NH_3_-N (mg/100 mL)	16.9	14.2	16.4	18.4	15.3	2.23

* Different from the control (CTL, 0 mg/L); *p* ≤ 0.05. ^a^ Control (0 mg/L). ^b^ Monensin (10 mg/L). ^c^ BCVFA: Branched-chain VFA = iso-butyrate + iso-valerate.

**Table 7 animals-13-03280-t007:** Effect of roasted pellets biochar on rumen microbial fermentation and CH_4_ production in vitro.

			Biochar (% of DM)			
Item	CTL ^a^	MON ^b^	1%	2%	5%	SEM
Gas production (mL)	46.7	35.9 *	50.1	48.1	49.5	1.27
CH_4_ (mL)	8.80	6.51 *	8.99	8.79	8.82	0.136
pH	6.61	6.64	6.58	6.60	6.60	0.037
Total VFA (m*M*)	87.3	78.5	78.7	86.0	84.6	7.20
VFA (mol/100 mol)						
Acetate (A)	62.7	60.6 *	62.7	62.6	62.5	0.46
Propionate (P)	20.0	23.8 *	20.0	20.1	20.0	0.30
Butyrate	11.9	10.4 *	11.9	11.9	11.9	0.21
Valerate	1.74	2.03 *	1.72	1.75	1.77	0.053
BCVFA ^c^	3.64	3.20 *	3.66	3.70	3.72	0.101
A:P	3.13	2.56 *	3.13	3.12	3.12	0.053
NH_3_-N (mg/100 mL)	16.9	14.2	16.2	13.2	15.3	1.86

* Different from the control (CTL, 0 mg/L); *p* ≤ 0.05. ^a^ Control (0 mg/L). ^b^ Monensin (10 mg/L). ^c^ BCVFA: Branched-chain VFA = iso-butyrate + iso-valerate.

**Table 8 animals-13-03280-t008:** Effect of Local-R biochar on rumen microbial fermentation and CH_4_ production in vitro.

			Biochar (% of DM)			
Item	CTL ^a^	MON ^b^	1%	2%	5%	SEM
Gas production (mL)	46.7	35.9 *	48.6	49.2	49.6	1.04
CH_4_ (mL)	8.80	6.51 *	8.77	8.46	8.63	0.147
pH	6.61	6.64	6.61	6.58	6.59	0.032
Total VFA (m*M*)	87.3	78.5	76.5	82.2	81.8	6.65
VFA (mol/100 mol)						
Acetate (A)	62.7	60.6 *	62.9	62.5	62.3	0.50
Propionate (P)	20.0	23.8 *	20.0	20.1	20.1	0.31
Butyrate	11.9	10.4 *	11.8	11.9	12.0	0.20
Valerate	1.74	2.03 *	1.70	1.75	1.78	0.058
BCVFA ^c^	3.64	3.20 *	3.63	3.71	3.78	0.111
A:P	3.13	2.56 *	3.14	3.11	3.10	0.058
NH_3_-N (mg/100 mL)	16.9	14.2	13.6	17.2	14.3	1.48

* Different from the control (CTL, 0 mg/L); *p* ≤ 0.05. ^a^ Control (0 mg/L). ^b^ Monensin (10 mg/L). ^c^ BCVFA: Branched-chain VFA = iso-butyrate + iso-valerate.

**Table 9 animals-13-03280-t009:** Effect of Local-L biochar on rumen microbial fermentation and CH_4_ production in vitro.

			Biochar (% of DM)			
Item	CTL ^a^	MON ^b^	1%	2%	5%	SEM
Gas production (mL)	46.7	35.9 *	50.2	49.2	50.4	1.13
CH_4_ (mL)	8.80	6.51 *	8.88	8.90	8.84	0.126
pH	6.61	6.64	6.59	6.61	6.58	0.034
Total VFA (m*M*)	87.3	78.5	77.9	82.7	85.8	7.55
VFA (mol/100 mol)						
Acetate (A)	62.7	60.6 *	62.1	62.7	62.5	0.57
Propionate (P)	20.0	23.8 *	20.1	20.0	20.0	0.34
Butyrate	11.9	10.4 *	12.2	11.8	12.0	0.23
Valerate	1.74	2.03 *	1.81	1.73	1.75	0.062
BCVFA ^c^	3.64	3.20 *	3.89	3.71	3.73	0.114
A:P	3.13	2.56 *	3.09	3.13	3.13	0.069
NH_3_-N (mg/100 mL)	16.9	14.2	14.0	14.3	18.6	1.52

* Different from the control (CTL, 0 mg/L); *p* ≤ 0.05. ^a^ Control (0 mg/L). ^b^ Monensin (10 mg/L). ^c^ BCVFA: Branched-chain VFA = iso-butyrate + iso-valerate.

## Data Availability

All data for this study are available within the manuscript.

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
