# Peer review of "Assessment of the Effects of Commercial or Locally Engineered Biochars Produced from Different Biomass Sources and Differing in Their Physical and Chemical Properties on Rumen Fermentation and Methane Production In Vitro"

_animals, 2023, doi:10.3390/ani13203280_

Round 1
Reviewer 1 Report
The study aimed to assess the effects of seven biochars (four commercial and three locally engineered) on rumen microbial fermentation and methane production.
Although no effect was observed, the study is of scientific value as a lot of physicochemical measurements were taken to investigate their effects on the ruminal fermentation process and it clearly demonstrates that biochar is at least until the present time no reliable option to mitigate enteric methane production.
However, I see some weaknesses in the statistical evaluation of the data and the presentation of the results.
The presentation of results is far too long and can be shortened very much. Since none of the biochars used in the present study influenced the measured values, I think this can be described very quickly in one paragraph and does not need to be explained separately for each product.
Furthermore, I would really recommend using a two-factor model for the statistical evaluation of the data including the interaction between product and dosing and to show the results in one table as the values for control and monensin are repeated 7 times.
L 27 Can you give some concrete examples of these properties that can be suspected here
L121 Why did you use a ratio of 1:5? As I know a ratio of 1:2 is recommended. Can you please explain?
L147-149 Why were these dosages chosen? Please add an explanation to the text.
L164 Please give the total amount of substrate (TMR) which was weighted into one bottle.
L173-174 What was the total amount of substrate incubated in one bottle? Was the gas production corrected by blanks and the amount of substrate?
L165-166 Did you include standard feedstuff with know gas production? This is highly recommended for these kinds of experiments.
L200 As already mentioned in the beginning I wonder, why you did not take the inclusion rate into account in your statistical modell? I think you can evaluate the experiment with the effects product, dosage and the interaction between product and dosage. This would also improve the data presentation as I would recommend showing all the data in one big table. Because the data for control and monensin are, now, repeated in every table for each product which somehow artificially drags out the entire presentation.
L215 Happy not Hammy
L220-222 Shouldn't be total carbon higher than fixed carbon? Can you please explain that to me?
L246 should be 5 not 3%
L248 What was the intension for showing the pH-value? From my point of view comparing pH in a buffered in vitro system makes no sense since these systems should be designed to keep the pH-value almost constant over the entire test period and to buffer the fermentation end products (VFA) quickly and continuously.
L264 should be 5 not 3%
L265-266 In table 4, although only small difference, the CH4 production is marked significant with 2% inclusion rate.
L280 should be 5 not 3%
L299 should be 5 not 3%
L303 double negative...
L413 This might be true regarding the physicochemcial characteristics that you evaluated. but there maybe others....?
L418.419 Redundant. You should be careful not to repeat the same arguments over and over again. I think the discussion needs to be improved in this respect.
L427 Redundant. Line 403-404
